# OpenReview forum: "Bi-perspective Splitting Defense: Achieving Clean-Data-Free Backdoor Security"
_ICLR.cc/2025/Conference — Submitted to ICLR 2025_

### Official Review · Reviewer_PNes · 2024-10-27

**Soundness:** 3
**Presentation:** 3
**Contribution:** 3
**Rating:** 6
**Confidence:** 4

**Summary:**

The paper discusses a method to defend against backdoor attacks in deep neural networks (DNNs) by training reliable models from poisoned datasets, addressing the challenge of lacking additional clean data due to privacy concerns or scarcity. It proposes a Bi-perspective Splitting Defense (BSD) that uses open set recognition-based splitting (OSS) and altruistic model-based data splitting (ALS) to divide the dataset effectively, initializing a pool of clean samples. BSD also employs class completion and selective dropping to prevent class underfitting and backdoor overfitting. Experiments on three benchmark datasets and against seven attacks show that BSD improves Defense Effectiveness Rating (DER) by an average of 16.29% compared to five state-of-the-art defenses, while maintaining high Clean Accuracy (CA) and managing Attack Success Rate (ASR) effectively, thus providing robust security without needing extra clean data.

**Strengths:**

The strengths of the proposed Bi-perspective Splitting Defense (BSD) method highlighted in the paper include:

1. **Efficiency in Utilizing Available Data**: BSD makes effective use of target labels and identifies clean samples from the poisoned dataset, reducing the dependency on additional clean data, which might be scarce or raise privacy concerns.

2. **Mitigation Strategies**: BSD incorporates class completion and selective dropping, helping to avoid issues like class underfitting and backdoor overfitting that could otherwise degrade model performance.

3. **Balanced Performance**: While enhancing security, BSD maintains high Clean Accuracy (CA) and manages Attack Success Rate (ASR), ensuring that the model remains accurate on clean data while defending against adversarial inputs.

**Weaknesses:**

The weaknesses identified in the paper are outlined as follows:

1. **Dependency on Target Recognition**: A significant limitation of the proposed methodology lies in its dependence on accurate target recognition. This dependency introduces a vulnerability; if an advanced attack manages to circumvent the existing detection mechanisms, the entire method could become ineffective. In my opinion, the superior performance is predicated on the assumption that the target class can be accurately identified.   Additionally, the framework's effectiveness diminishes when dealing with complex scenarios such as backdoor attacks that involve multiple targets, including those with dual-target labels or all-to-all attack configurations.

2. **Absence of Comparative Analysis with Relevant Work**: The proposed dual-model training framework, which leverages an auxiliary model to support the primary model, shares similarities with the approach detailed in "[1]". However, the manuscript lacks a comparative analysis of this work. I see the proposed method has many differences from [1], but incorporating such a comparison would significantly bolster the credibility and relevance of the current submission by providing a clearer differentiation and understanding of the advantages and limitations relative to existing methodologies.

Reference:
[1] The Victim and The Beneficiary: Exploiting a Poisoned Model to Train a Clean Model on Poisoned Data, ICCV 2023 Oral paper

**Questions:**

The authors have already provided comprehensive analyses and experiments.

However, an unresolved question remains: How effectively can the proposed method be applied to scenarios involving multi-target attacks?

---

> ### Author Response · Authors · 2024-11-25
> **R1: Response to Weakness 1 & Question 1(Dependency on Target Recognition and multi-target attacks)**
>
> Thank you for raising this insightful concern regarding the dependency on accurate target recognition and multi-target attacks.
>
> **Dependency on Target Recognition:**
>
> - As demonstrated in ABL **([2], Figure 7, page 16)**, local gradients accent achieves a detection rate of over 50% for poisoned samples across all tested attacks. This indicates that $y_t$ can reliably be predicted—a result also supported by our experiments.
>
>   Furthermore, we presented an alternative **energy-based estimation method** in **Appendix E.3** (Appendix D.3 in the first submission), which achieves a 100% $y_t$ identification rate under various attacks, including BadNet, Blended, WaNet, and LC (**Figure 11**) ***(It should be noted that this alternative method can directly replace the original pseudo-target identification function without modifying the training process or increasing overhead)***. The dual validation provided by these two methods makes it robust against potential unseen failures.
>
>   Additionally, in additional experiments where attacker chosed different target labels, we verified that the prediction accuracy of pseudo-targets remained 100%, as shown in **Table 10 (Appendix A.4)**, where ASR values are consistently low under all settings. Meanwhile, we tested the $y_t$ approximation on different target labels (the first 10 classes) on GTSRB in **Table12 (Appendix A.6)** as well.
>
>   Moreover, as mentioned on **page 5, line 268**, we operate under a relatively weak assumption that is broadly applicable in common scenarios:
>
>   > In common scenarios where the dataset is a large but well-known benchmark dataset, the number of samples in each class is known to the public, and $y_t$ can be approximated through label statistics.
>
> **Multi-target attacks:**
>
> - Multi-target attacks (commonly all-to-all attacks) indeed pose a certain level of risk to our defense mechanism, particularly in the OSR and selective drop components. However, we would like to note that **all2all attacks are not typically considered essential scenarios in backdoor defense research currently [1-5]**, for the following reasons:
>
>   1. The increased number of trigger-target pairs in all2all attacks requires significantly more training epochs for success. And all2all attacks reduce clean accuracy and exhibit slower convergence, making them easier to detect. ([6], Page 2: “As the number of classes increases, the accuracy and the attack success rate will decrease.”)
>   2. Research on all2all attacks remains limited ([7], Page 10: “However, there were only a few studies on all-to-all attacks. How to better design the all-to-all attack and the analysis of its properties remain blank.”).
>   3. In practical applications, all2all attacks do not allow attackers to arbitrarily control predictions to specific targets, limiting their real-world threat.
>
>   Nevertheless, we still conducted supplementary experiments on BadNets with an all2all setting to address your concerns **(Appendix A.5)**.
>
>   - **Attack setting**: Following BadNets, with $y_t = (y+1) % n_c$, where $n_c$ is the number of classes.
>   - **Defense setting**: To handle multiple target labels, BSD incurs additional computational costs by iterating through all classes as pseudo-targets during OSS. Clean indices from each pseudo-target are intersected to form the final OSS result. Additionally, we early stop at Stage 2 to avoid meaningless cost in Stage 3.
>
>   Since all-to-all attacks do not fundamentally change the nature of poison-label attacks, OSS remains effective for each individual classes. We visualized OSS spliting results in **Figure 5 (Appendix A.5)**, which reveals effective separation of clean samples of OSS. The CA, ASR, and DER performance are presented in **Table 11 (Appendix A.5)**, demonstrating a significant DER improvement compared to baseline methods. Notably, while BSD’s ASR increases under all-to-all attacks, it effectively limits the attack success rate to the level of random prediction ($1/n_c = 10%$).
>
>   In conclusion, our BSD method remains effective against all-to-all attacks. Furthermore, the OSS module can serve as a highly effective component for identifying clean samples in other backdoor defense methods.
>
> We hope these clarifications and additional results address your concern.

---

> ### Author Response · Authors · 2024-11-25
> **R1 (Cont.)**
>
> *Copies of Table 10, Table 11, Table 12 are provided below for ease of reference (as Openreview doesn't support inserting figures, Figure 5 is only accessible in the revised manuscript):*
>
> **Table 10.** The clean accuracy (CA%), attack success rate (ASR%), and defense effective rating (DER%) of our BSD against 3 representitive backdoor attacks with different target labels on CIFAR-10.
> |            | <------ | BadNet  | ------> | <------ | blended | ------> | <------ |  WaNet  | ------> |
> | :--------: | :-----: | :-----: | :-----: | :-----: | :-----: | :-----: | :-----: | :-----: | :-----: |
> | **Target** | **CA**  | **ASR** | **DER** | **CA**  | **ASR** | **DER** | **CA**  | **ASR** | **DER** |
> |     0      |  95.0   |   0.8   |  99.6   |  95.0   |   0.4   |  99.0   |  91.9   |   0.3   |  99.0   |
> |     1      |  94.9   |   0.5   |  99.8   |  94.9   |   0.5   |  98.9   |  94.2   |   0.3   |  99.8   |
> |     2      |  95.1   |   0.8   |  99.6   |  94.7   |   0.9   |  98.7   |  90.9   |   0.7   |  98.3   |
> |     3      |  95.1   |   0.9   |  99.6   |  94.9   |   0.8   |  98.8   |  94.5   |   0.8   |  99.6   |
> |     4      |  95.0   |   0.2   |  99.9   |  94.8   |   0.6   |  98.9   |  92.4   |   0.2   |  99.3   |
> |     5      |  95.1   |   1.7   |  99.2   |  95.0   |   0.5   |  98.9   |  91.9   |   0.4   |  98.9   |
> |     6      |  95.1   |   0.6   |  99.7   |  93.9   |   0.6   |  98.8   |  92.6   |   0.3   |  99.3   |
> |     7      |  94.7   |   0.3   |  99.8   |  92.6   |   0.4   |  98.2   |  90.3   |   0.0   |  98.3   |
> |     8      |  92.0   |   0.3   |  98.4   |  95.1   |   0.5   |  98.9   |  91.8   |   0.3   |  98.9   |
> |     9      |  94.9   |   0.3   |  99.8   |  94.0   |   0.4   |  98.9   |  94.0   |   0.2   |  99.9   |
> |    Avg     |  94.7   |   0.6   |  99.5   |  94.5   |   0.6   |  98.8   |  92.5   |   0.4   |  99.1   |
>
> **Table 11.** The clean accuracy (CA%), attack success rate (ASR%), and defense effective rating (DER%) of ASD and our BSD against BadNets-all2all on CIFAR-10.
>
> |             | <------ | BadNets-all2all | ------> |
> | :---------: | :-----: | :-------------: | :-----: |
> | **Method**  | **CA**  |     **ASR**     | **DER** |
> | No  Defense |  91.8   |      93.8       |    -    |
> |     ASD     |  70.2   |       2.4       |  84.9   |
> | BSD (Ours)  |  91.2   |      10.5       |  91.3   |
>
> **Table 12.** Testing the $y_t$ approximation on different target labels (the first 10 classes) on GTSRB.
>
> | Attack  | 0    | 1    | 2    | 3    | 4    | 5    | 6    | 7    | 8    | 9    |
> | ------- | ---- | ---- | ---- | ---- | ---- | ---- | ---- | ---- | ---- | ---- |
> | BadNets | ✓    | ✓    | ✓    | ✓    | ✓    | ✓    | ✓    | ✓    | ✓    | ✓    |
> | Blended | ✓    | ✓    | ✓    | ✓    | ✓    | ✓    | ✓    | ✓    | ✓    | ✓    |
> | LC      | ✓    | ✓    | ✓    | ✓    | ✓    | ✓    | ✓    | ✓    | ✓    | ✓    |

---

> ### Author Response · Authors · 2024-11-25
> **R2: Response to Weakness 2 (More baselines)**
>
> Thank you for pointing out the valuable baseline models,VaB [8]. The victim model in VaB is somewhat similar to our altruistic model. However there exist many difference between VaB and our BSD: 1) VaB split the dataset using loss related statistics only, while BSD includes sementic-based spliting using open set recognition (OSS); 2) VaB focus on improving the semi-supervised training itself with the proposed AttentionMix, while our BSD focus on better splitting, which means AttentionMix could be a potential module that could further improve BSD.
>
>   We added two recent defense, VaB [8] and D-ST/D-BR [9]. The additional baselines are implemented based on the official implementation. We use CIFAR-10 as the dataset. Since the label-consistent attack is not consistently implemented, we use SIG as a clean label attack here. The results of these comparisons are presented in **Table 7 (Appendix A.2)**, which further demonstrates the accuracy and robustness of the proposed metrics.
>
>   We hope this addition addresses your concern and strengthens the experimental validation of our approach.
>
>   ----
>
>   *A copy of Table 7 is provided below for ease of reference:*
>
>   **Table 7.** The clean accuracy (CA%), attack success rate (ASR%), and defense effective rating (DER%) of 2 additional backdoor defense methods and our BSD against 4 threatening backdoor attacks on CIFAR-10. The best and second best results are in **bold** andunderlined.
>
>   |            | <------     | Badnets    | ------>     | <------     | Blended    | ------>     | <------     | WaNet      | ------>     | <------     | SIG        | ------>     | Avg         | TIME     |
>   | :--------: | :---------: | :--------: | :---------: | :---------: | :--------: | :---------: | :---------: | :--------: | :---------: | :---------: | :--------: | :---------: | :---------: | :------: |
>   | **Method** | **CA**      | **ASR**    | **DER**     | **CA**      | **ASR**    | **DER**     | **CA**      | **ASR**    | **DER**     | **CA**      | **ASR**    | **DER**     | **DER**     | **Cost** |
>   | VaB        |94.0 | 1.3        |98.9 |94.2 |1.1 |98.6 |93.6 |1.7 |99.1 |94.0 | 66.6       |63.8 |90.1 | 5.5      |
>   | D-ST       | 66.8        | 5.7        | 83.1        | 65.0        | 7.1        | 81.1        | 60.8        | 15.2       | 76.0        | 87.9        | 95.1       | 46.5        | 71.6        | 4.3      |
>   | D-BR       | 87.5        | **0.8**    | 95.9        | 83.0        | 80.7       | 53.2        | 16.9        | 14.6       | 54.3        | 85.7        |0.1 | 92.9        | 74.1        | -        |
>   | BSD(Ours)  | **95.1**    |0.9 | **99.6**    | **94.9**    | **0.8**    | **98.8**    | **94.5**    | **0.8**    | **99.6**    | **93.8**    | **0.0**    | **97.0**    | **98.7**    | **3.2**  |

---

> ### Author Response · Authors · 2024-11-25
> **Reference(s):**
>
> \[1\]: *Huang K, Li Y, Wu B, et al. (2022)* Backdoor defense via decoupling the training process[J].
>
>   \[2\]: *Li Y, Lyu X, Koren N, et al.(2021)* Anti-backdoor learning: Training clean models on poisoned data[J].
>
>   \[3\]: *Zhu Z, Wang R, Zou C, et al. (2023)* The Victim and The Beneficiary: Exploiting a Poisoned Model to Train a Clean Model on Poisoned Data[C].
>
>   \[4\]: *Guan J, Liang J, He R. (2024)* Backdoor Defense via Test-Time Detecting and Repairing[C].
>
>   \[5\]: *Zhang Z, Liu Q, Wang Z, et al. (2023)* Backdoor defense via deconfounded representation learning[C].
>
>   \[6\]: *Huang B, Lok J C, Liu C, et al. [2024]* Poisoning-based Backdoor Attacks for Arbitrary Target Label with Positive Triggers[J].
>
>   \[7\]: *Li Y, Jiang Y, Li Z, et al. (2022)* Backdoor learning: A survey[J].
>
>   \[8\]: *Zhu Z, Wang R, Zou C, et al. (2023)* The Victim and The Beneficiary: Exploiting a Poisoned Model to Train a Clean Model on Poisoned Data[C].
>
>   \[9\]: *Chen, W., Wu, B., & Wang, H. (2022).* Effective backdoor defense by exploiting sensitivity of poisoned samples.

---

> > ### Comment · Reviewer_PNes · 2024-11-26
> > **Response**
> >
> > Most of my concerns are addressed. So, I raised my score.

---

> > > ### Author Response · Authors · 2024-11-26
> > >
> > > Thank you for your positive feedback and for raising your score. We greatly appreciate your thoughtful review and constructive suggestions, which have helped us improve the quality of our work.

---

### Official Review · Reviewer_5XZg · 2024-10-27

**Soundness:** 3
**Presentation:** 2
**Contribution:** 2
**Rating:** 3
**Confidence:** 5

**Summary:**

This paper proposes an in-training clean-data-free backdoor defense method where the defender is required to train a clean model from scratch given a poisoned dataset without the need of any additional clean data. The key to success is to distinguish between clean samples and poisoned samples. To this end, they propose a novel identification mechanism which involves two main procedures. The first procedure is initializing a pool of clean samples and a pool of poisoned samples based on open set recognition-based splitting and altruistic model-based splitting. The second procedure is improving these pools with class completion and selective dropping strategy.

**Strengths:**

1. The proposed defense does not require any extra clean data.

2. They compare the problem of identifying clean target samples from poisoned samples with the open set recognition-based splitting problem of identifying UUCs from UKCs; inspired by which, they propose the identification mechanism.

**Weaknesses:**

1. Limited applied scenarios: The proposed defense is limited to backdoor attacks with a single target class. This is because the first step in identification involves identifying the single target class from other classes; however, in some popular attacks like BadNet’s all-to-all attacks, all classes are target classes, which disables the proposed defense.

2. The proposed method seems a bit complex as it includes two main steps—pool initialization and pool updating—and each step further involves two sub-steps, respectively. These steps aim to distinguish samples from different perspectives. Only after the total four steps, the poisoned samples are filtered out from clean samples. A complex mechanism is totally fine; but, there are two related issues: 1) does the necessity of pool updating validates that the effectiveness of pool initialization is not very good? 2) it seems that the effectiveness of each step highly depends on the performance of previous steps, e.g., if the identification of the target class is wrong, then all subsequent steps are useless. That is to say, accumulative errors may exist in the proposed method and which could lead to bad defense performance.

**Questions:**

1. In the exp setup, it’s reported to use 7 backdoor attacks while only 3 of them are presented in Table 1. That is to say, the performance against the remaining 4 attacks on the three benchmark datasets is not shown in the paper.

2. Considering the given threat model (i.e., in-training clean-data-free defense), it seems that sota defenses D-ST&D-BR [1] could also serve as baselines. These methods leverage the sensitivity of poisoned samples to transformations, which is quite different from the semantics and losses used in this paper; thus, it would be interesting to compare them and show that the proposed metrics are more accurate.

3. Is the accuracy of identifying the target class reported in the paper? What if the identified class is wrong, will it affect the performance of the following steps?

4. A type in Line 83: “first initialize” -> “first initializes”

[1] Chen, W., Wu, B., & Wang, H. (2022). Effective backdoor defense by exploiting sensitivity of poisoned samples. Advances in Neural Information Processing Systems, 35, 9727-9737.

---

> ### Author Response · Authors · 2024-11-25
> **R1: Response to Weakness 1 (Defending against all-to-all attacks)**
>
> Thank you for highlighting this potential threat. All-to-all (all2all) attacks may pose challenges to certain components of our defense, particularly OSS and selective drop. However, we would like to note that **all2all attacks are not typically considered essential scenarios in backdoor defense research currently [1-5]**, for the following reasons:
>
> 1. The increased number of trigger-target pairs in all2all attacks requires significantly more training epochs for success. And all2all attacks reduce clean accuracy and exhibit slower convergence, making them easier to detect. ([6], Page 2: “As the number of classes increases, the accuracy and the attack success rate will decrease.”)
>
>   2. Research on all2all attacks remains limited ([7], Page 10: “However, there were only a few studies on all-to-all attacks. How to better design the all-to-all attack and the analysis of its properties remain blank.”).
>   3. In practical applications, all2all attacks do not allow attackers to arbitrarily control predictions to specific targets, limiting their real-world threat.
>
> Nevertheless, we still conducted supplementary experiments on BadNets with an all2all setting to address your concerns **(Appendix A.5)**.
>
> - **Attack setting**: Following BadNets, with $y_t = (y+1) % n_c$, where $n_c$ is the number of classes.
>
> - **Defense setting**: To handle multiple target labels, BSD incurs additional computational costs by iterating through all classes as pseudo-targets during OSS. Clean indices from each pseudo-target are intersected to form the final OSS result. Additionally, we early stop at Stage 2 to avoid meaningless costs in Stage 3.
>
> Since all-to-all attacks do not fundamentally change the nature of poison-label attacks, OSS remains effective for each individual class. We visualized OSS splitting results in **Figure 5 (Appendix A.5)**, which reveals the effective separation of clean samples of OSS. The CA, ASR, and DER performance are presented in **Table 11 (Appendix A.5)**, demonstrating a significant DER improvement compared to baseline methods. Notably, while BSD’s ASR increases under all-to-all attacks, it effectively limits the attack success rate to the level of random prediction ($1/n_c = 10%$).
>
> In conclusion, our BSD method remains effective against all-to-all attacks. Furthermore, the OSS module can serve as a highly effective component for identifying clean samples in other backdoor defense methods.
>
> We hope these clarifications and additional results address your concern.
>
> ---
>
> *A copy of Table 11 is provided below for ease of reference (as Openreview doesn't support inserting figures, Figure 5 is only accessible in the revised manuscript):*
>
> **Table 11.** The clean accuracy (CA%), attack success rate (ASR%), and defense effective rating (DER%) of ASD and our BSD against BadNets-all2all on CIFAR-10.
>
> |             | <------ | BadNets-all2all | ------> |
> | :---------: | :-----: | :-------------: | :-----: |
> | **Method**  | **CA**  |     **ASR**     | **DER** |
> | No  Defense |  91.8   |      93.8       |    -    |
> |     ASD     |  70.2   |       2.4       |  84.9   |
> | BSD (Ours)  |  91.2   |      10.5       |  91.3   |

---

> > ### Comment · Reviewer_5XZg · 2024-11-27
> > **Performance against all-to-all attacks**
> >
> > Yes, it seems that it takes more computational costs for BSD to defend against all-to-all backdoor attacks, as it needs to iterate through all classes as pseudo-targets during OSS. But it's good to know that BSD is capable of defending against such attacks, although ASR remains a bit high. Lastly, I believe that all-to-all backdoor attack still serves as a good attack baseline to demonstrate a defense method's effectiveness, as many top conferences papers like [1,2] still involve this baseline.
> > [1] Dongxian Wu, Yisen Wang: Adversarial Neuron Pruning Purifies Backdoored Deep Models. NeurIPS 2021: 16913-16925
> > [2] Yige Li, Xixiang Lyu, Nodens Koren, Lingjuan Lyu, Bo Li, Xingjun Ma: Neural Attention Distillation: Erasing Backdoor Triggers from Deep Neural Networks. ICLR 2021

---

> ### Author Response · Authors · 2024-11-25
> **R2: Response to Weakness 2 & Question 3 (Pool updating related issues)**
>
> Thank you for your insightful comments and constructive feedback. First, we would like to emphasize that our proposed method is not overly complex. The primary contribution lies in the **pool initialization phase**, which employs two lightweight, complementary methods to filter backdoored samples from the training set. Importantly, these methods do not rely on clean seed samples and require minimal computational resources, making them well-suited for the challenging settings of backdoor defenses. This phase significantly enhances the effectiveness of existing data-splitting-based backdoor defense approaches. Regarding the subsequent **pool updating phase**, it involves relatively minor adjustments to address two specific issues encountered in prior works: class underfitting and backdoor overfitting. Below, we provide detailed responses to the two concerns you raised in **R2.1 & R2.2**.

---

> ### Author Response · Authors · 2024-11-25
> **R2.1: Weakness 2, issue 1**
>
> We respectfully disagree with the claim that the effectiveness of our pool initialization is suboptimal. On the contrary, our clean pool initialization demonstrates strong performance, with high precision and a reasonably sized initial clean pool. This ensures that **even without our pool updating modifications**, the model’s performance does not degrade significantly **(Table 5, Section 5.7)**. Below, we explain the rationale for introducing pool updating and elaborate on why our clean pool initialization achieves good performance.
>
> - **Rationale for Pool Updating:**
>
>   Regardless of whether pool initialization is conducted without clean seed samples (as in our method) or aided by a small number of clean seed samples (as in ASD and similar methods), the defender’s capacity is inherently constrained. It is infeasible for the defender to directly establish a **large and secure clean pool** directly. Consequently, the size of the initialized clean pool is relatively small, necessitating pool updating to expand the clean pool and further improve model performance.
>
> - **Reasons for the Strong Performance of Our Clean Pool Initialization:**
>
>   Existing works (e.g., ASD Stage 2 [8], DBD Stage 3 [1]) rely on **adaptive updating based solely on loss values** to partition clean and poisoned samples into separate pools. However, this approach can lead to two forms of cascading errors that ultimately cause model collapse:
>
>   1. **Class Underfitting:** If samples from certain classes are absent in the clean pool, their loss values will increase, making it harder for those samples to be included in the clean pool in subsequent updates. This results in poor classification accuracy for those classes.
>   2. **Backdoor Overfitting:** If poisoned samples (false positives) are mistakenly added to the clean pool, their loss values decrease. This, in turn, makes it easier for other poisoned samples with the same trigger to enter the clean pool, exacerbating the problem.
>
>   In contrast, our pool initialization employs a **bi-perspective intersection strategy** to select clean samples, significantly reducing false positives and achieving higher precision. This mitigates the risk of backdoor overfitting. Additionally, our initial clean pool is not excessively small, ensuring better coverage across all categories and reducing the likelihood of class underfitting.

---

> ### Author Response · Authors · 2024-11-25
> **R2.2: Weakness 2, issue 2 & Question 3**
>
> The core issue raised in Weakness 2, Issue 2 concerns the accuracy of identifying $y_t$, which impacts both the OSR and selective drop mechanisms. However, as demonstrated in ABL **([2], Figure 7, page 16)**, local gradients accent achieves a detection rate of over 50% for poisoned samples across all tested attacks. This indicates that $y_t$ can reliably be predicted—a result also supported by our experiments.
>
> Furthermore, we presented an alternative **energy-based estimation method** in **Appendix E.3** (Appendix D.3 in the first submission), which achieves a 100% $y_t$ identification rate under various attacks, including BadNet, Blended, WaNet, and LC (**Figure 11**) ***(It should be noted that this alternative method can directly replace the original pseudo-target identification function without modifying the training process or increasing overhead)***. The dual validation provided by these two methods makes it robust against potential unseen failures.
>
> Additionally, in additional experiments where attacker chosed different target labels, we verified that the prediction accuracy of pseudo-targets remained 100%, as shown in **Table 10 (Appendix A.4)**, where ASR values are consistently low under all settings. Meanwhile, we tested the $y_t$ approximation on different target labels (the first 10 classes) on GTSRB in **Table12 (Appendix A.6)** as well.
>
> Moreover, as mentioned on **page 5, line 268**, we operate under a relatively weak assumption that is broadly applicable in common scenarios:
>
> > In common scenarios where the dataset is a large but well-known benchmark dataset, the number of samples in each class is known to the public, and $y_t$ can be approximated through label statistics.
>
> Regarding the concern about **accumulative errors or incorrect class identification**, we have not encountered significant issues in our experiments. To explore this interesting question further, we conducted additional tests by **forcing** $y_t$ **to be assigned to an incorrect class** and observed the model’s performance  **(Figure 6, Appendix A.6)**.
>
> ----
>
> *Copies of Table 10 and Table 12 are provided below for ease of reference (as Openreview doesn't suppot inserting figures, Figure 6 is only accessible in the revised manuscript):*
>
> **Table 10.** The clean accuracy (CA%), attack success rate (ASR%), and defense effective rating (DER%) of our BSD against 3 representitive backdoor attacks with different target labels on CIFAR-10.
>
> |            | <------ | BadNet  | ------> | <------ | blended | ------> | <------ |  WaNet  | ------> |
> | :--------: | :-----: | :-----: | :-----: | :-----: | :-----: | :-----: | :-----: | :-----: | :-----: |
> | **Target** | **CA**  | **ASR** | **DER** | **CA**  | **ASR** | **DER** | **CA**  | **ASR** | **DER** |
> |     0      |  95.0   |   0.8   |  99.6   |  95.0   |   0.4   |  99.0   |  91.9   |   0.3   |  99.0   |
> |     1      |  94.9   |   0.5   |  99.8   |  94.9   |   0.5   |  98.9   |  94.2   |   0.3   |  99.8   |
> |     2      |  95.1   |   0.8   |  99.6   |  94.7   |   0.9   |  98.7   |  90.9   |   0.7   |  98.3   |
> |     3      |  95.1   |   0.9   |  99.6   |  94.9   |   0.8   |  98.8   |  94.5   |   0.8   |  99.6   |
> |     4      |  95.0   |   0.2   |  99.9   |  94.8   |   0.6   |  98.9   |  92.4   |   0.2   |  99.3   |
> |     5      |  95.1   |   1.7   |  99.2   |  95.0   |   0.5   |  98.9   |  91.9   |   0.4   |  98.9   |
> |     6      |  95.1   |   0.6   |  99.7   |  93.9   |   0.6   |  98.8   |  92.6   |   0.3   |  99.3   |
> |     7      |  94.7   |   0.3   |  99.8   |  92.6   |   0.4   |  98.2   |  90.3   |   0.0   |  98.3   |
> |     8      |  92.0   |   0.3   |  98.4   |  95.1   |   0.5   |  98.9   |  91.8   |   0.3   |  98.9   |
> |     9      |  94.9   |   0.3   |  99.8   |  94.0   |   0.4   |  98.9   |  94.0   |   0.2   |  99.9   |
> |    Avg     |  94.7   |   0.6   |  99.5   |  94.5   |   0.6   |  98.8   |  92.5   |   0.4   |  99.1   |
>
> **Table 12.** Testing the $y_t$ approximation on different target labels (the first 10 classes) on GTSRB.
>
> | Attack  | 0    | 1    | 2    | 3    | 4    | 5    | 6    | 7    | 8    | 9    |
> | ------- | ---- | ---- | ---- | ---- | ---- | ---- | ---- | ---- | ---- | ---- |
> | BadNets | ✓    | ✓    | ✓    | ✓    | ✓    | ✓    | ✓    | ✓    | ✓    | ✓    |
> | Blended | ✓    | ✓    | ✓    | ✓    | ✓    | ✓    | ✓    | ✓    | ✓    | ✓    |
> | LC      | ✓    | ✓    | ✓    | ✓    | ✓    | ✓    | ✓    | ✓    | ✓    | ✓    |

---

> > ### Comment · Reviewer_5XZg · 2024-11-27
> > **Accuracy of identifying correct target class**
> >
> > Q1: Thanks for mentioning the existing method ABL has a high detection rate, and your alternative method presented in Appendix has a high identification rate. But how about the identification rate for the original method presented in the main paper? I may miss these results and it would be appreciated if you could point it out. Thanks!
> >
> > Q2: According to the new Figure 6, it seems that even if the identified class is wrong, the proposed method still achieves high ACC and low ASR. That is to say, it seems that the proposed method works in most time even with wrongly detected target classes. Could you give more explanations on this? as I feel a bit hard to understand this; moreover, if this is true, what is the end for identifying correct target class as an initial step? Look forward to your reply in understanding these, thanks!

---

> ### Author Response · Authors · 2024-11-25
> **R3: Response to Question 1 (Clarification on experiment results)**
>
> We apologize for any confusion caused by the presentation of our results. Due to space constraints and time limitations, **Table 1** does not include all possible combinations of attacks, defenses, and datasets. However, the performance results for the remaining four attacks that you are concerned about are presented in **Table 2**.

---

> > ### Comment · Reviewer_5XZg · 2024-11-27
> > **Incomplete results (missing results on the other two benchmark datasets)**
> >
> > Thanks for the reply. Yes, I knew the performance on the remaining four attacks on **CIFAR-10** were shown in Table 2. But I am wondering if the performance on the remaining four attacks on **the other two benchmark datasets** are missing? Thanks!

---

> ### Author Response · Authors · 2024-11-25
> **R4: Response to Question 2 (More baselines)**
>
> Thank you for pointing out the valuable baseline models, D-ST & D-BR [9]. We agree that this method, which leverage the sensitivity of poisoned samples to transformations, offer an interesting comparison given their distinct approach from the semantics and losses used in our method.
>
>   We added two recent defense, D-ST/D-BR [9] and VaB [10]. The additional baselines are implemented based on the official implementation. We use CIFAR-10 as the dataset. Since the label-consistent attack is not consistently implemented, we use SIG as a clean label attack here. The results of these comparisons are presented in **Table 7 (Appendix A.2)**, which further demonstrates the accuracy and robustness of the proposed metrics.
>
>   We hope this addition addresses your concern and strengthens the experimental validation of our approach.
>
>   ----
>
>   *A copy of Table 7 is provided below for ease of reference:*
>
>   **Table 7.** The clean accuracy (CA%), attack success rate (ASR%), and defense effective rating (DER%) of 2 additional backdoor defense methods and our BSD against 4 threatening backdoor attacks on CIFAR-10. The best results are in **bold**.
>
>   |            | <------     | Badnets    | ------>     | <------     | Blended    | ------>     | <------     | WaNet      | ------>     | <------     | SIG        | ------>     | Avg         | TIME     |
>   | :--------: | :---------: | :--------: | :---------: | :---------: | :--------: | :---------: | :---------: | :--------: | :---------: | :---------: | :--------: | :---------: | :---------: | :------: |
>   | **Method** | **CA**      | **ASR**    | **DER**     | **CA**      | **ASR**    | **DER**     | **CA**      | **ASR**    | **DER**     | **CA**      | **ASR**    | **DER**     | **DER**     | **Cost** |
>   | VaB        |94.0 | 1.3        |98.9 |94.2 |1.1 |98.6 |93.6 |1.7 |99.1 |94.0 | 66.6       |63.8 |90.1 | 5.5      |
>   | D-ST       | 66.8        | 5.7        | 83.1        | 65.0        | 7.1        | 81.1        | 60.8        | 15.2       | 76.0        | 87.9        | 95.1       | 46.5        | 71.6        | 4.3      |
>   | D-BR       | 87.5        | **0.8**    | 95.9        | 83.0        | 80.7       | 53.2        | 16.9        | 14.6       | 54.3        | 85.7        |0.1 | 92.9        | 74.1        | -        |
>   | BSD(Ours)  | **95.1**    |0.9 | **99.6**    | **94.9**    | **0.8**    | **98.8**    | **94.5**    | **0.8**    | **99.6**    | **93.8**    | **0.0**    | **97.0**    | **98.7**    | **3.2**  |

---

> > ### Comment · Reviewer_5XZg · 2024-11-27
> >
> > Thanks for updating these results.

---

> ### Author Response · Authors · 2024-11-25
> **R5: Response to Question 4 (typo errors)**
>
> Thank you for your attention to detail in identifying this typographical error. We have corrected “first initialize” to “first initializes” in the latest version of the manuscript.

---

> ### Author Response · Authors · 2024-11-25
> **Reference(s):**
>
> \[1\]: *Huang K, Li Y, Wu B, et al. (2022)* Backdoor defense via decoupling the training process[J].
>
> \[2\]: *Li Y, Lyu X, Koren N, et al.(2021)* Anti-backdoor learning: Training clean models on poisoned data[J].
>
> \[3\]: *Zhu Z, Wang R, Zou C, et al. (2023)* The Victim and The Beneficiary: Exploiting a Poisoned Model to Train a Clean Model on Poisoned Data[C].
>
> \[4\]: *Guan J, Liang J, He R. (2024)* Backdoor Defense via Test-Time Detecting and Repairing[C].
>
> \[5\]: *Zhang Z, Liu Q, Wang Z, et al. (2023)* Backdoor defense via deconfounded representation learning[C].
>
> \[6\]: *Huang B, Lok J C, Liu C, et al. [2024]* Poisoning-based Backdoor Attacks for Arbitrary Target Label with Positive Triggers[J].
>
> \[7\]: *Li Y, Jiang Y, Li Z, et al. (2022)* Backdoor learning: A survey[J].
>
> \[8\]: *Gao K, Bai Y, Gu J, et al. (2023)* Backdoor defense via adaptively splitting poisoned dataset.
>
> \[9\]: *Chen, W., Wu, B., & Wang, H. (2022).* Effective backdoor defense by exploiting sensitivity of poisoned samples.
>
> \[10\]: *Zhu Z, Wang R, Zou C, et al. (2023)* The Victim and The Beneficiary: Exploiting a Poisoned Model to Train a Clean Model on Poisoned Data[C].

---

> ### Author Response · Authors · 2024-11-28
> **Follow-up response to reviewer 5XZg**
>
> # Target class identification
>
> Thank you for your thoughtful questions and for taking the time to review our work. Below are our detailed responses to your follow-up questions:
>
> **Re Q1:**
>
> The original method used in our main paper is derived from ABL (local gradient ascent), which you have noted that they already verified the performance. Specifically, it achieves a detection rate of over 50% for poisoned samples across various tested attacks (which means the majority label must be the target label). Based on these findings, we adopted this method in the early stages of our work, where experiments with a fixed target label (set consistently to label 3) showed that the method worked adequately. Since it works well, we did not specifically log its approximation result (we are sorry about that). Therefore, in the previous response, we only mentioned it as "a result also supported by our experiments."
>
> > as demonstrated in ABL **([2], Figure 7, page 16)**, local gradients accent achieves a detection rate of over 50% for poisoned samples across all tested attacks. This indicates that $y_t$ can reliably be predicted—a result also supported by our experiments.
>
> However, when we expanded the evaluation to include different target labels, we actually observed that the original method was not stable enough under all scenarios. For instance, in the same setting as Table 12, it works well against Badnets and LC, but failed in gtsrb-blended-target 0, gtsrb-blended-target 6, and gtsrb-blended-target 8 (i.e. it only achieved success rate of approximately $27/30 = 90\%$).
>
> Considering that our alternative method neither affects other parts of the pipeline nor adds to the training overhead, and as you can see, we have thoroughly validated it, we currently find it more suitable as the main approach for $y_t$ approximation. In future versions, we may reposition the alternative method as the main approach.
>
> It should be noted that the mentioned two methods for approximating $y_t$ are neither the major contribution of our paper nor the only approach (e.g., existing methods in literature [10]&[11]). And in the worst cases (if encountered), we have the last resort that turns to a relatively weak assumption that is broadly applicable in common scenarios:
>
> > In common scenarios where the dataset is a large but well-known benchmark dataset, the number of samples in each class is known to the public, and $y_t$ can be approximated through label statistics.
>
> **Re Q2:**
>
> The key reason for this phenomenon lies in the behavior of certain attacks, particularly BadNets (all-to-one). This type of attack does not heavily rely on semantic alignment but can be effectively mitigated only using loss statistics.
>
> This specific experiment of Figure 6 was conducted on CIFAR-BadNets. In this setup, BadNets achieves a high ASR quickly, where even when the main model collapses, the loss of poisoned samples on the altruistic model drops significantly faster than on the main model, causing significant loss discrepancy. This discrepancy increases also because the altruistic model is initialized and trained on a different portion of the poisoned dataset, plus that the initialization of the altruistic model is performed on the entire poisoned dataset. Thus, as subsequent training progresses, the loss-discrepancy-based splitting mechanism gradually separates the two types of samples, reducing the presence of poisoned samples in the clean pool to an ideal state.
>
> **It is important to emphasize that we are not proving that $y_t$ is unimportant or useless.** This experiment just explores the robustness of the model under incorrect target class identification, which is an interesting question you raised. The robustness observed in Figure 6 is specific to attacks like BadNets, where loss statistics alone suffice for effective splitting. However, as we highlighted in **page 18, lines 930-941**, the defense is not always effective under incorrect $y_t$, particularly for broader attack scenarios or more complex dataset settings. This reinforces the necessity of the bi-perspective splitting mechanism, where $y_t$ is used in OSS module for robust defense.
>
> > when faced with broader attack scenarios and dataset settings, relying solely on loss statistics may not be sufficient to ensure effective defense. **Fortunately**, our experiments demonstrate the strong robustness of the proposed Pseudo Target Approximation method. The OSS mechanism functioned as expected, enabling a resilient bi-perspective defense under challenging conditions.
>
> -----
>
>
> \[10\]: *Guo J, Li A, Liu C. (2022)* Aeva: Black-box backdoor detection using adversarial extreme value analysis.
>
> \[11\]: *Ma W, Wang D, Sun R, et al. (2022)* The" Beatrix''Resurrections: Robust Backdoor Detection via Gram Matrices.

---

> > ### Author Response · Authors · 2024-11-28
> > **Follow-up response to reviewer 5XZg (Cont.)**
> >
> > # **All-to-all attacks**
> >
> > Thank you for your insightful observation and constructive feedback. We appreciate your recognition of BSD’s capability to defend against all-to-all backdoor attacks. We regret that, due to time constraints, the adaption we made was not sufficient to achieve the robust performance you might have expected against all-to-all attacks.
> >
> > However, we believe that the current results meet a ‘passable’ level of effectiveness. More importantly, the OSS module’s ability to isolate clean samples (as illustrated in Figure 5) holds significant potential for augmenting existing defense methods. This may add to the robustness of other approaches when dealing with such challenging attack scenarios.
> >
> > We respect your perspective that the all-to-all backdoor attack remains an essential baseline for evaluating the effectiveness of defense methods. Moving forward, we will pay attention to this issue in our future research.
> >
> > ---
> >
> > # More results on benchmark datasets
> >
> > Thank you for your valuable comment and for pointing out this important concern. We sincerely appreciate your attention to the comprehensiveness of our evaluation.
> >
> > In the initial experimental design, we referred to the setup of DBD (ICLR 2022), which primarily presented results on three representative attacks (i.e., BadNets, Blended, and WaNet) across CIFAR-10 and ImageNet in the main text. Building upon this, our main table further incorporated results on the GTSRB dataset. As a result, we mistakenly assumed that this setup was sufficient. And due to time constraints (considering the number of baseline methods and attacks), we did not evaluate all additional attacks across all datasets. This oversight has now been brought to our attention, and we sincerely apologize for not involving the dataset-attack combinations you are particularly interested in.
> >
> > We have revised the abstract to remove potentially misleading descriptions of our experimental setup.
> >
> > Furthermore, while the remaining time of the discussion phase prevents us from fully expanding our experiments across all dataset-attack combinations, we have conducted additional evaluations to include the results of the remaining four attacks on the GTSRB dataset. These results are presented below.
> >
> > |        | <------ |  LC   | ------> | <------ | SIG  | ------> | <------ | ReFool | ------> | <------ | Narcissus | ------> | avgDER |
> > | :----: | :-----: | :---: | :-----: | :-----: | :--: | :-----: | :-----: | :----: | :-----: | :-----: | :-------: | :-----: | :----: |
> > | Method |   CA    |  ASR  |   DER   |   CA    | ASR  |   DER   |   CA    |  ASR   |   DER   |   CA    |    ASR    |   DER   |        |
> > |  Non   |  97.3   | 100.0 |    -    |  97.1   | 97.1 |    -    |  97.5   |  99.8  |    -    |  97.3   |   94.5    |    -    |   -    |
> > |  BSD   |  97.5   |  0.1  |  100.0  |  95.7   | 8.9  |  93.4   |  96.1   |  0.3   |  99.1   |  94.0   |    0.0    |  95.6   |  97.0  |
> >
> > ---
> >
> > ---
> >
> > **Thank you again for your valuable and constructive feedback！**

---

> ### Comment · Reviewer_5XZg · 2024-11-28
> **Summary for the rebuttal**
>
> Thanks for the authors’ efforts in rebuttal, but considering the followings, I will remain my score.
>
> - The proposed method is restricted to defending against limited attack scenarios, all-to-one attacks in particular. In rebuttal, the authors show the capability of defending against more complex attacks, like all-to-all attacks; but, the attack performance is not quite satisfactory (up to 10% ASR); the authors indicated that better performance could be achieved if using more fine-grained adaptation specifically for the attack, which is usually not allowed for the defender though.
>
> - The authors admitted that the experimental results were not complete and appeared to not align with the description in the paper (they claimed they will modify this later in the paper), as they reported the results against BadNet, Blended, and WaNet on three benchmark datasets while only reporting the results against LC, SIG, ReFool, and NarCISSUS on the CIFAR-10 dataset (they claimed they will add these in the future).
>
> - In terms of the accuracy of detecting correct target class, the authors firstly refered to ABL [1] and explained ABL has reported high detection rate; secondly refered to an alternative method introduced in Appendix which shows high detection rate. As no results for the original method proposed in the main paper are exhibited, I further kindly asked about it in the 2nd round of rebuttal. The authors admitted that the orginal method has an unstable detection rate for different target classes, and they will exchange the positions of the original method and the alternative method later, which seems to be a great change to the submitted paper though.
>
>
> [1] Li, Y., Lyu, X., Koren, N., Lyu, L., Li, B., & Ma, X. (2021). Anti-backdoor learning: Training clean models on poisoned data. Advances in Neural Information Processing Systems, 34, 14900-14912.

---

### Official Review · Reviewer_Rm4s · 2024-10-28

**Soundness:** 3
**Presentation:** 3
**Contribution:** 3
**Rating:** 6
**Confidence:** 4

**Summary:**

Backdoor attacks threaten deep neural networks (DNNs) by embedding hidden vulnerabilities through data poisoning. While researchers have explored training benign models from poisoned data, effective defenses often rely on additional clean datasets, which are challenging to obtain due to privacy concerns and data scarcity.

To tackle these issues, the paper proposes the Bi-perspective Splitting Defense (BSD), which splits the dataset based on semantic and loss statistics using open set recognition (OSS) and altruistic model-based data splitting (ALS). This approach enhances clean pool initialization and includes strategies to prevent class underfitting and backdoor overfitting.

Extensive experiments on three benchmark datasets against seven attacks show that BSD is robust, achieving an average 16.29% improvement in Defense Effectiveness Rating (DER) compared to five state-of-the-art defenses, while maintaining minimal compromise in Clean Accuracy (CA) and Attack Success Rate (ASR).

**Strengths:**

- This paper explores a novel method to defend backdoor attacks which does not rely on clean subsets of data.

- The proposed method is simple but effective and the good performance obtained by the experiments strongly supports this point.

- The ablation study is organized well to clearly demonstrate the whole proposed method. And it makes the paper easy to follow.

**Weaknesses:**

- I wonder **why the proposed BSD can effectively resist the clean-label backdoor attacks.** Clean-label backdoor attacks manipulate samples from the target class while keeping their labels unchanged. Since the BSD framework formulates backdoor defense within a semi-supervised learning context, it seems that whether the clean-label poisoned samples are part of the labeled or unlabeled subset, **the model can still associate the trigger with the target label in clean-label backdoor attacks.** Therefore, I am curious about how BSD manages to effectively resist the three clean-label attacks demonstrated in Table 2.

- I recommend conducting further experiments to assess whether BSD can successfully defend against backdoor attacks **with different target labels.** This could provide valuable insights into the robustness of the defense mechanism across various scenarios.

- Typos: #Line 848 --- #Line 849, 5*10e-4.

**Questions:**

Listed in the weakness of the paper.

Score can be improved if concerns listed above are resolved.

---

> ### Author Response · Authors · 2024-11-25
> **R1: Response to Weakness 1 (Discussion on resistance to clean-label attack)**
>
> Thank you for pointing out this interesting topic. We focus on this phenomenon in the early stage of our research as well. **To put the conclusion first, the primary reason BSD can effectively resist clean-label backdoor attacks lies in how the MixMatch algorithm processes unlabeled data.** Specifically, MixMatch applies a mixup operation that visually weakens the trigger and prevents it from being directly associated with the target label. For further clarity, we have included visualizations of MixMatch’s effect in **Appendix A.3**. Additionally, clean-label attacks inherently have lower attack success rates, and as long as the poisoned samples are correctly placed into the poison pool (unlabeled data), BSD effectively mitigates their impact.
>
> During the early design phase of BSD, we explored a module using diffusion models for attack-agnostic image purification, combining inpainting and denoising techniques in reverse diffusion to remove both local triggers (local pixel modification like badnets) and global triggers (global modification like blended). While the module achieved visually satisfactory purification, they had limited success in improving defense performance against clean-label attacks, as trigger removal often degraded overall image quality.
>
> To better understand this phenomenon, we analyzed related semi-supervised defense methods like ASD [1] and DBD [2]. These works report good performance against label-consistent attacks (e.g., ASD [1] has an ASR near 0% against LC attack). DBD [2] included a brief explanation of its effectiveness against clean-label attacks (**[2] Appendix O, page 25**), but they focus on explaining why clean-label poison samples can be separated instead of explaining why the semi-supervised learning in the last stage works.
>
> Through the reproduction of ASD, we observed that in successful cases, poison samples were correctly classified into the poison pool, while failures often occurred when poison samples remained in the clean pool. For instance, we reviewed log files from two failed ASD experiments and found that a significant number of poison samples were misclassified into the clean pool during the final training rounds **(Table 8, Appendix A.3)**. The number of poison samples in clean pool exceeds 500 in average (poisoning ratio 2% in the clean pool, close to a random split), which is the main reason for the failed run.
>
> This led us to hypothesize that the key factor influencing clean-label attack defense is still the accurate classification of poison samples into the poison pool. Further investigation revealed that the mixup operation in MixMatch plays a critical role. MixMatch not only removes original labels but also mixes multiple inputs on the image level, effectively weakening the visual impact of the trigger. The operation is described as follows:
>
> $$ \tilde{x} = \lambda' x_l + (1 - \lambda') x_u, \quad \tilde{y} = \lambda' y_l + (1 - \lambda') y_u, \quad\lambda\sim Beta(\alpha,\alpha),\quad \lambda'=\text{max}(\lambda,1-\lambda). $$
>
> Because $\alpha=0.75$ in the default setting, $\lambda'$ has a mathematical expectation of approximately 0.78, the mixed inputs diminish the prominence of the trigger in $x_u$. We visualized the mixed samples together with the argmaxed mixed label in **Figure 4 (Appendix A.3)** to better present the mixup operation. Furthermore, we follow ASD to set a 5 times smaller $\lambda_u$ value (15 vs. MixMatch’s recommended 75), reducing the influence of unlabeled data and further mitigating clean-label attacks.
>
> Finally, to verify this hypothesis, we manually enforced a secure clean-poison split where no poison samples were included in the clean pool. Under this condition, MixMatch effectively nullified the impact of clean-label attacks, as shown in our new experimental results **(Table 9, Appendix A.3)**.

---

> ### Author Response · Authors · 2024-11-25
> **R1 (Cont.)**
>
> *Copies of Table 8, Table 9 are provided below for ease of reference (as Openreview doesn't support inserting figures, Figure 4 is only accessible in the revised manuscript):*
>
>   **Table 8.** Number of poison samples in the clean pool of failed runs of ASD (CIFAR-10, LC attack, poisoning ratio 2.5%). The number of poison samples in the clean pool exceeds 500 on average (poisoning ratio 2% in the clean pool), which is the main reason that it fails to resist LC attack.
>
>   | Epoch                        | 111  | 112  | 113  | 114  | 115  | 116  | 117  | 118  | 119  | 120  | mean  | std   |
>   | ---------------------------- | ---- | ---- | ---- | ---- | ---- | ---- | ---- | ---- | ---- | ---- | ----- | ----- |
>   | Num of poison (failed run 1) | 405  | 567  | 683  | 321  | 695  | 280  | 303  | 595  | 391  | 910  | 515   | 208.4 |
>   | Num of poison (failed run 2) | 936  | 466  | 838  | 544  | 658  | 146  | 435  | 370  | 530  | 345  | 526.8 | 234.9 |
>
>   **Table 9.** The clean accuracy (CA%), attack success rate (ASR%), and defense effective rating (DER%) of the model trained on secured clean pool against clean-label attacks.
>
>   |            | <------ | LC-CIFAR | ------> | <------ | LC-GTSRB | ------> | <------ | SIG-CIFAR | ------> | <------ | Narcissus-CIFAR | ------> |
> | :--------: | :-----: | :------: | :-----: | :-----: | :------: | :-----: | :-----: | :-------: | :-----: | :-----: | :-------------: | :-----: |
> | **Method** | **CA**  | **ASR**  | **DER** | **CA**  | **ASR**  | **DER** | **CA**  |  **ASR**  | **DER** | **CA**  |     **ASR**     | **DER** |
> | No defense |  94.4   |   99.9   |    -    |  97.3   |  100.0   |    -    |  95.0   |   95.2    |    -    |  95.2   |      99.5       |    -    |
> | MixMatch*  |  94.2   |   0.0    |  99.9   |  97.4   |   0.0    |  100.0  |  94.6   |    0.0    |  97.4   |  94.5   |       0.0       |  99.3   |

---

> ### Author Response · Authors · 2024-11-25
> **R2: Response to Weakness 2 (Try on different target labels)**
>
> Thank you for your suggestion. In response, we have conducted additional experiments on CIFAR-10, evaluating all target labels across the three primary attack types **(Table 10, Appendix A.4)**. The results are basically consistent with those observed when the target label was set to 3, further confirming the robustness of BSD across different target label scenarios.
>
> ---
>
> *A copy of Table 10 is provided below for ease of reference:*
>
> **Table 10.** The clean accuracy (CA%), attack success rate (ASR%), and defense effective rating (DER%) of our BSD against 3 representitive backdoor attacks with different target labels on CIFAR-10.
> |            | <------ | BadNet  | ------> | <------ | blended | ------> | <------ |  WaNet  | ------> |
> | :--------: | :-----: | :-----: | :-----: | :-----: | :-----: | :-----: | :-----: | :-----: | :-----: |
> | **Target** | **CA**  | **ASR** | **DER** | **CA**  | **ASR** | **DER** | **CA**  | **ASR** | **DER** |
> |     0      |  95.0   |   0.8   |  99.6   |  95.0   |   0.4   |  99.0   |  91.9   |   0.3   |  99.0   |
> |     1      |  94.9   |   0.5   |  99.8   |  94.9   |   0.5   |  98.9   |  94.2   |   0.3   |  99.8   |
> |     2      |  95.1   |   0.8   |  99.6   |  94.7   |   0.9   |  98.7   |  90.9   |   0.7   |  98.3   |
> |     3      |  95.1   |   0.9   |  99.6   |  94.9   |   0.8   |  98.8   |  94.5   |   0.8   |  99.6   |
> |     4      |  95.0   |   0.2   |  99.9   |  94.8   |   0.6   |  98.9   |  92.4   |   0.2   |  99.3   |
> |     5      |  95.1   |   1.7   |  99.2   |  95.0   |   0.5   |  98.9   |  91.9   |   0.4   |  98.9   |
> |     6      |  95.1   |   0.6   |  99.7   |  93.9   |   0.6   |  98.8   |  92.6   |   0.3   |  99.3   |
> |     7      |  94.7   |   0.3   |  99.8   |  92.6   |   0.4   |  98.2   |  90.3   |   0.0   |  98.3   |
> |     8      |  92.0   |   0.3   |  98.4   |  95.1   |   0.5   |  98.9   |  91.8   |   0.3   |  98.9   |
> |     9      |  94.9   |   0.3   |  99.8   |  94.0   |   0.4   |  98.9   |  94.0   |   0.2   |  99.9   |
> |    Avg     |  94.7   |   0.6   |  99.5   |  94.5   |   0.6   |  98.8   |  92.5   |   0.4   |  99.1   |

---

> ### Author Response · Authors · 2024-11-25
> **R3: Response to Weakness 3 (typo errors)**
>
> Thank you for your attention to detail in identifying this typographical error. We have corrected "$5\times10^-4$" to "$5\times10^{-4}$" in the latest version of the manuscript.

---

> ### Author Response · Authors · 2024-11-25
> **Reference(s):**
>
> \[1\]: *Gao K, Bai Y, Gu J, et al. (2023)* Backdoor defense via adaptively splitting poisoned dataset.
>
> \[2\]: *Huang K, Li Y, Wu B, et al. (2022)* Backdoor defense via decoupling the training process[J].

---

> > ### Comment · Reviewer_Rm4s · 2024-11-26
> >
> > I appreciate the authors' response, which solves my concerns. I have raised my score.

---

> > > ### Author Response · Authors · 2024-11-26
> > >
> > > Thank you for your kind response and for raising your score. We sincerely appreciate your constructive feedback, which has been invaluable in enhancing our work.

---

### Official Review · Reviewer_HqAA · 2024-11-01

**Soundness:** 3
**Presentation:** 4
**Contribution:** 3
**Rating:** 6
**Confidence:** 4

**Summary:**

This paper addresses the challenge of defending deep neural networks against backdoor attacks in the absence of clean data subsets. It introduces a novel defense mechanism called Bi-perspective Splitting Defense (BSD) that uses semantic and loss statistics characteristics for dataset splitting. The approach involves two innovative initial pool splitting techniques, Open Set Recognition-based Splitting (OSS) and Altruistic Model-based Splitting (ALS), and it enhances defense through subsequent updates of class completion and selective dropping strategies. The method demonstrates substantial improvements over state-of-the-art defenses across multiple benchmarks.

**Strengths:**

1. The paper introduces a novel method for defending against backdoor attacks without the need for clean data, addressing a significant limitation in previous methods.
2. Extensive experiments across multiple datasets and attack scenarios demonstrate the robustness and effectiveness of the proposed method, outperforming several state-of-the-art defenses.
3. The paper details multiple defensive strategies that contribute to its effectiveness, such as class completion and selective dropping, which are well-integrated into the defense strategy.

**Weaknesses:**

1. The complexity of the proposed method involving multiple models and sophisticated data splitting strategies could be a barrier to practical deployment and computational efficiency.
2. While the method is empirically successful, the paper could be improved by providing deeper theoretical insights into why the specific strategies employed are effective.
3. The effectiveness of the method might depend on specific neural network architectures, and its adaptability to different or future architectures is not fully addressed.
4. The paper could benefit from a more detailed discussion on scenarios where BSD might fail or be less effective, which would be crucial for practical applications and future improvements.

**Questions:**

See weaknesses above.

---

> ### Author Response · Authors · 2024-11-25
> **R1: Response to Weakness 1 (Concerns on method complexity)**
>
> Thank you for pointing out the perceived complexity of our proposed method. While our approach may initially seem intricate, it is, in fact, conceptually straightforward and practical in implementation.
>
>   The core idea of our method is simple: leveraging two complementary, non-clean-seed-involved, and lightweight pool initialization methods to filter backdoored samples from the training set. This enhances existing data splitting-based backdoor defenses under challenging scenarios. Specifically, the two pool initialization methods—semantic-based OSS (open set recognition-based splitting) and loss statistics-based ALS (altruistic model-based data splitting)—are innovative. They utilize open set recognition and altruistic models, respectively, to accurately identify clean data within the training set. This establishes a robust foundation to prevent model collapse during training.
>
>   The subsequent strategies, are minor modifications to the pool update process based on existing works, play a relatively smaller role in overall performance improvement. This is particularly true in datasets with a large number of categories, where their impact is less significant.
>
>   Regarding computational efficiency, as analyzed in **Section 5.6 (page 9–10, lines 481–495)**, our method demonstrates comparable or even superior computational costs relative to state-of-the-art methods like ASD [1]. In addition, when adding more baselines, we recorded the time consumption of two more competitive defenses on CIFAR-10 **(Table 7, Appendix A.2)**, where our BSD still has less cost. This is due to few updates required by the altruistic model, the use of standard supervised losses for altruistic model parameter updates, and a more balanced pool size. Moreover, our approach is significantly more efficient than highly complex methods such as DBD, **offering a favorable balance between performance and computational overhead.**
>
>   ---
> *A copy of Table 7 is provided below for ease of reference:*
>
> **Table 7.** The clean accuracy (CA%), attack success rate (ASR%), and defense effective rating (DER%) of 2 additional backdoor defense methods and our BSD against 4 threatening backdoor attacks on CIFAR-10. The best results are in **bold**.
>
>   |            | <------     | Badnets    | ------>     | <------     | Blended    | ------>     | <------     | WaNet      | ------>     | <------     | SIG        | ------>     | Avg         | TIME     |
>   | :--------: | :---------: | :--------: | :---------: | :---------: | :--------: | :---------: | :---------: | :--------: | :---------: | :---------: | :--------: | :---------: | :---------: | :------: |
>   | **Method** | **CA**      | **ASR**    | **DER**     | **CA**      | **ASR**    | **DER**     | **CA**      | **ASR**    | **DER**     | **CA**      | **ASR**    | **DER**     | **DER**     | **Cost** |
>   | VaB        |94.0 | 1.3        |98.9 |94.2 |1.1 |98.6 |93.6 |1.7 |99.1 |94.0 | 66.6       |63.8 |90.1 | 5.5      |
>   | D-ST       | 66.8        | 5.7        | 83.1        | 65.0        | 7.1        | 81.1        | 60.8        | 15.2       | 76.0        | 87.9        | 95.1       | 46.5        | 71.6        | 4.3      |
>   | D-BR       | 87.5        | **0.8**    | 95.9        | 83.0        | 80.7       | 53.2        | 16.9        | 14.6       | 54.3        | 85.7        |0.1 | 92.9        | 74.1        | -        |
>   | BSD(Ours)  | **95.1**    |0.9 | **99.6**    | **94.9**    | **0.8**    | **98.8**    | **94.5**    | **0.8**    | **99.6**    | **93.8**    | **0.0**    | **97.0**    | **98.7**    | **3.2**  |

---

> ### Author Response · Authors · 2024-11-25
> **R2: Response to Weakness 2 (Suggestion on theoretical insights)**
>
> We appreciate the reviewer’s suggestion to provide deeper theoretical insights into the effectiveness of the specific strategies employed. However, we would like to clarify that our work does not fall under the category of certified backdoor defenses, which are typically designed with theoretical guarantees to ensure robustness under certain assumptions. Consequently, a theoretical analysis is not a necessary or standard component for evaluating the effectiveness of our approach.
>
> It is important to note that certified backdoor defenses, while theoretically robust, often demonstrate limited empirical performance in practice due to the restrictive assumptions underlying their guarantees **([2], Section 2.2, Page 3, Line 18-19)**. Most empirically-driven backdoor defense methods, including those closely related to our work, such as ASD[1], DBD[2], and ABL[3], also do not provide theoretical analyses. This supports the rationale for our focus on empirical validation and suggests that the lack of a theoretical analysis does not constitute a weakness in our work.
>
> That said, if theoretical insights are of interest, several components of our BSD framework align with existing theoretical studies in the literature. For instance:
>
> - The module addressing open-set recognition is theoretically grounded in works such as [4].
> - Our alternative estimation method for $y_t$ has theoretical connections to energy statistics, discussed in [5].
>
> We hope this clarifies our position and the relevance of theoretical analyses within the context of our work.

---

> ### Author Response · Authors · 2024-11-25
> **R3: Response to Weakness 3 (Robustness to model architectures)**
>
> Thank you for raising the concern regarding the adaptability of our method to different or future neural network architectures. We appreciate the opportunity to clarify this aspect of our work.
>
> As noted in **Section 5.4** of the manuscript (page 8-9, line 427-431), BSD makes no assumptions about specific model structures, ensuring its compatibility and versatility across various architectures. To validate this, we conducted experiments on MobileNet-V2, demonstrating its effectiveness beyond the primary architectures evaluated **(Table 3, Section 5.4)**. Note that model structures of baseline methods were changed accordingly as well, ensuring a fair and consistent evaluation.
> In summary, BSD’s design ensures its robustness to different neural network architectures.

---

> ### Author Response · Authors · 2024-11-25
> **R4: Response to Weakness 4 (Expanding Discussion on potential failures)**
>
> Thank you for suggesting a more detailed discussion on scenarios where BSD might fail or be less effective. We have already addressed many potential issues, including clean label attacks (Section 5.3), model structures (Section 5.4), datasets (Section 5.2), poisoning rates (Section 5.5), parameter sensitivity (Section 5.7), no-attack scenarios (Appendix E.4), and adaptive attacks (Appendix E.5).
>
> In response to your feedback, we have expanded our discussion by including experiments on all2all attacks and different target labels, further clarifying the limitations and applicability of BSD.
>
> - Different target labels:
>
>   Thank you for your suggestion. In response, we have conducted additional experiments on CIFAR-10, evaluating all target labels across the three primary attack types **(Table 10, Appendix A.4)**. The results are basically consistent with those observed when the target label was set to 3, further confirming the robustness of BSD across different target label scenarios.
>
>
> - All2all attack：
>
>     All-to-all (all2all) attacks may pose challenges to certain components of our defense, particularly OSS and selective drop. However, we would like to note that **all2all attacks are not typically considered essential scenarios in backdoor defense research currently [1-5]**. Nevertheless, we still conducted supplementary experiments on BadNets with an all2all setting to address reviewers' concerns **(Appendix A.5)**.
>
>     - **Attack setting**: Following BadNets, with $y_t = (y+1) % n_c$, where $n_c$ is the number of classes.
>
>     - **Defense setting**: To handle multiple target labels, BSD incurs additional computational costs by iterating through all classes as pseudo-targets during OSS. Clean indices from each pseudo-target are intersected to form the final OSS result. Additionally, we early stop at Stage 2 to avoid meaningless costs in Stage 3.
>
>     Since all-to-all attacks do not fundamentally change the nature of poison-label attacks, OSS remains effective for each individual class. We visualized OSS splitting results in **Figure 5 (Appendix A.5)**, which reveals the effective separation of clean samples of OSS. The CA, ASR, and DER performance are presented in **Table 11 (Appendix A.5)**, demonstrating a significant DER improvement compared to baseline methods. Notably, while BSD’s ASR increases under all-to-all attacks, it effectively limits the attack success rate to the level of random prediction ($1/n_c = 10%$).
>
>     In conclusion, our BSD method remains effective against all-to-all attacks. Furthermore, the OSS module can serve as a highly effective component for identifying clean samples in other backdoor defense methods.
>
>
>
> ---
>
>   *Copies of Table 10 and Table 11 are provided below for ease of reference:*
>
>   **Table 10.** The clean accuracy (CA%), attack success rate (ASR%), and defense effective rating (DER%) of our BSD against 3 representitive backdoor attacks with different target labels on CIFAR-10.
>
> |            | <------ | BadNet  | ------> | <------ | blended | ------> | <------ |  WaNet  | ------> |
> | :--------: | :-----: | :-----: | :-----: | :-----: | :-----: | :-----: | :-----: | :-----: | :-----: |
> | **Target** | **CA**  | **ASR** | **DER** | **CA**  | **ASR** | **DER** | **CA**  | **ASR** | **DER** |
> |     0      |  95.0   |   0.8   |  99.6   |  95.0   |   0.4   |  99.0   |  91.9   |   0.3   |  99.0   |
> |     1      |  94.9   |   0.5   |  99.8   |  94.9   |   0.5   |  98.9   |  94.2   |   0.3   |  99.8   |
> |     2      |  95.1   |   0.8   |  99.6   |  94.7   |   0.9   |  98.7   |  90.9   |   0.7   |  98.3   |
> |     3      |  95.1   |   0.9   |  99.6   |  94.9   |   0.8   |  98.8   |  94.5   |   0.8   |  99.6   |
> |     4      |  95.0   |   0.2   |  99.9   |  94.8   |   0.6   |  98.9   |  92.4   |   0.2   |  99.3   |
> |     5      |  95.1   |   1.7   |  99.2   |  95.0   |   0.5   |  98.9   |  91.9   |   0.4   |  98.9   |
> |     6      |  95.1   |   0.6   |  99.7   |  93.9   |   0.6   |  98.8   |  92.6   |   0.3   |  99.3   |
> |     7      |  94.7   |   0.3   |  99.8   |  92.6   |   0.4   |  98.2   |  90.3   |   0.0   |  98.3   |
> |     8      |  92.0   |   0.3   |  98.4   |  95.1   |   0.5   |  98.9   |  91.8   |   0.3   |  98.9   |
> |     9      |  94.9   |   0.3   |  99.8   |  94.0   |   0.4   |  98.9   |  94.0   |   0.2   |  99.9   |
> |    Avg     |  94.7   |   0.6   |  99.5   |  94.5   |   0.6   |  98.8   |  92.5   |   0.4   |  99.1   |
>
>   **Table 11.** The clean accuracy (CA%), attack success rate (ASR%), and defense effective rating (DER%) of ASD and our BSD against BadNets-all2all on CIFAR-10.
>
> |             | <------ | BadNets-all2all | ------> |
> | :---------: | :-----: | :-------------: | :-----: |
> | **Method**  | **CA**  |     **ASR**     | **DER** |
> | No  Defense |  91.8   |      93.8       |    -    |
> |     ASD     |  70.2   |       2.4       |  84.9   |
> | BSD (Ours)  |  91.2   |      10.5       |  91.3   |

---

> ### Author Response · Authors · 2024-11-25
> **Reference(s):**
>
> \[1\]: *Gao K, Bai Y, Gu J, et al. (2023)* Backdoor defense via adaptively splitting poisoned dataset.
>
> \[2\]: *Huang K, Li Y, Wu B, et al. (2022)* Backdoor defense via decoupling the training process[J].
>
> \[3\]: *Li Y, Lyu X, Koren N, et al.(2021)* Anti-backdoor learning: Training clean models on poisoned data[J].
>
> \[4\]: *Bendale A, Boult T E. (2016)* Towards open set deep networks[C].
>
> \[5\]: *Gao Y, Chen H, Sun P, et al. (2024)* Energy-based Backdoor Defense without Task-Specific Samples and Model Retraining[C].

---

> ### Comment · Reviewer_HqAA · 2024-11-26
> **Thanks for the response**
>
> Thanks for the response, I am basically satisfied so I  keep my score.

---

> > ### Author Response · Authors · 2024-11-26
> >
> > Thank you for your response and for taking the time to review our work. We appreciate your acknowledgment and your continued evaluation of our submission.

---

### Official Review · Reviewer_HMtE · 2024-11-05

**Soundness:** 3
**Presentation:** 3
**Contribution:** 3
**Rating:** 6
**Confidence:** 4

**Summary:**

This paper addresses the limitation of existing backdoor attack defenses, which typically require an auxiliary clean dataset that may be difficult to obtain. The authors propose a clean-data-free, end-to-end method to mitigate backdoor attacks. Their approach leverages two dynamically identified pools of data: one from open set recognition-based splitting and another from altruistic model-based splitting. These pools are then utilized in the main training loop, which the authors demonstrate to be effective in producing backdoor-free models, even when trained on poisoned datasets.

The authors validate their method using three benchmark datasets and test it against seven representative backdoor attacks, including both dirty-label and clean-label attacks. They compare their approach with five existing backdoor defenses and evaluate performance across two model architectures: ResNet-18 and MobileNet-v2.

**Strengths:**

- The paper targets a clear challenge in backdoor defense by eliminating the need for clean data, which is relevant to practical applications.

- The experimental evaluation is thorough, encompassing multiple datasets, attack types, and model architectures. The comparison against existing defense methods provides a meaningful context for the method's effectiveness.

- The authors conduct detailed ablation studies examining the impact of various hyperparameters, which helps in understanding the method's sensitivity and optimal configuration.

**Weaknesses:**

- The study primarily focuses on supervised image classification tasks. This limitation should be explicitly stated in both the abstract and introduction to better set reader expectations.

- The paper's motivation could be strengthened by acknowledging recent developments in clean data acquisition methods. Notable omissions include:

  *Zeng et al. (2023, Usenix Security)* on clean subset extraction from poisoned datasets

  *Pan et al. (2023, Usenix Security)* on backdoor data detection methods

  These omissions affect the paper's premise that clean data acquisition is inherently impossible.


- Technical Issues:

  A typographical error in line 331: "Mobileent-v2" should be "MobileNet-v2"

**References:**

*Zeng et al. (2023)*. "META-SIFT: How to Sift Out a Clean Data Subset in the Presence of Data Poisoning?" Usenix Security, 2023.

*Pan et al. (2023)*. "ASSET: Robust Backdoor Data Detection Across a Multiplicity of Deep Learning Paradigms." Usenix Security, 2023.

**Questions:**

See weakness.

---

> ### Author Response · Authors · 2024-11-25
> **R1: Response to Weakness 1 (Research focus claim)**
>
> Thank you for highlighting this concern. We have made the necessary revisions in the manuscript (page 1, line 44-45) to clarify the study's primary focus on supervised image classification tasks.
>
> **Meanwhile, I'd like to elaborate further regarding this limitation.** While our experiments indeed focus on image classification, it is important to note that our core modules, including open-set recognition and semi-supervised learning, are input-modality-agnostic. This enables transferability to tasks like text classification. For non-classification tasks such as object detection, the applicability may be partially limited due to the significant variations in backdoor attack implementations.  These valuable topics would be a meaningful part of our future work.

---

> ### Author Response · Authors · 2024-11-25
> **R2: Response to Weakness 2 (Expanding Discussion on Related Works)**
>
> We appreciate your insight in identifying these relevant studies. We have carefully considered the contributions of *Zeng et al. (2023)* [1] and *Pan et al. (2023)* [2] regarding clean subset extraction and backdoor detection. In general, these works indeed provide valuable insights into the poisoned data splitting problem and could inspire our future research. Accordingly, we have added them as extended related works. However, certain limitations make them unsuitable for the specific defense scenario addressed in our study.
>
> - *Zeng et al.* [1] proposed detecting poisoned data by identifying shifts in data distributions, which results in high prediction loss when training on the clean portion of a poisoned dataset and testing on the corrupted portion. They solve a relaxed of the splitting optimization problem with the help of a weight-assigning network. Although promising empirical results were presented, **the proposed META-SIFT only guarantees a relatively small subset ([1], page 10, Figure 5). As a result, META-SIFT still relies on effective downstream defenses, such as NAD and ASD, included in our baselines, while also increasing the hyperparameter search space.**
> - *Pan et al.* [2] are motivated by the same distributional shift phenomenon and proposed an effective splitting algorithm, ASSET. However, they **assume that the defender has an extra set of clean samples (named "base set" in [2])**, which doesn't suit the background of our paper, where no extra clean set is available. Plus, ASSET is faced with the same problem that **requires effective downstream defenses to conduct the defense.**
>
> **In summary,** these two works, along with other detection methods, do contribute to clean data splitting. However, they are faced with two major problems. 1) Cannot guarantee a 100% correct split that can be directly used for training; 2) Rely on an extra clean set which violates the constraints of our scenario, where clean data acquisition is infeasible due to privacy and cost concerns (page 2, lines 68–73). Thus, they do not challenge the premise of our study, and our proposed approach remains significant under the given constraints.

---

> ### Author Response · Authors · 2024-11-25
> **R3: Response to weakness 3 (typo errors)**
>
> Thank you for your attention to detail in identifying this typographical error. We have corrected "Mobileent-v2" to "MobileNet-v2" in the latest version of the manuscript.

---

> ### Author Response · Authors · 2024-11-25
> **Reference(s):**
>
> \[1\]: *Zeng et al. (2023)*. "META-SIFT: How to Sift Out a Clean Data Subset in the Presence of Data Poisoning?" Usenix Security, 2023.
>
> \[2\]: *Pan et al. (2023)*. "ASSET: Robust Backdoor Data Detection Across a Multiplicity of Deep Learning Paradigms." Usenix Security, 2023.

---

> ### Comment · Reviewer_HMtE · 2024-11-27
> **Thanks for response**
>
> Thank you for your detailed response. I would like to maintain my score of 6.

---

> > ### Author Response · Authors · 2024-11-27
> >
> > Thank you for taking the time to review our response and for your thoughtful consideration. We greatly appreciate your constructive feedback.

---

### Meta-Review · Area_Chair_Pj2t · 2024-12-20

**Metareview:**

This work proposed a novel training algorithm for defending against backdoor attack.

It received 4 detailed reviews. The strengths mentioned by reviewers include: the clean data free design, extensive evaluations on several attacks and datasets, extensive ablation studies.

Meanwhile, there are also several important concerns, mainly including:
1. Motivation: It is motivated that several backdoor defense methods require a clean subset, which is often difficult to acquire. But it seems that acquiring a small clean dataset is not difficult, which has been studied in existing works.
2. Effectiveness to all2all attack: Since the proposed method has to predict the target class, and the evaluation is only conducted on all2one attacks, the performance against all2all attacks is questionable, and is mentioned by several reviewers.
3. Complex overflow: there are several sequential steps, which may cause accumulated error. The resistance to some errors is questionable.
4. Robustness to different model architectures, and lack of theoretical insights.
5. Incomplete results in the presented tables.

There are sufficient discussions between authors and reviewers. I am happy to see that both the questions and answers are very professional. My judgements about above points as shown below:
1. I agree with the reviewer that acquiring a small clean dataset is not difficult. And, clean data free defenses have been widely developed in both pre-processing stage (i.e., poisoned data detection) and post training stage. What is the most relevant, most training based defense methods (the proposed method belongs to this category) also don't require clean data. Thus, the motivation is inappropriately claimed. Actually, it is not difficult to justify the importance of training based defense. I don't consider the inappropriate motivation is very significant, but it should not be considered as a strength.
2. The authors provided a few experiments on batnet-all2all, and showed that the ASR is still relatively high, and the cost is also high. It is obviously that the proposed method is not very suitable for all2all attacks. It is a significant limitation in practice.
3. This concern is very important and insightful. A complex design with multiple steps is likely to cause accumulated errors. This concern is not well addressed in the rebuttal.
4. When answering these two concerns, I feel that the authors are trying to changing the subject. It is claimed that the methodology design is independent with model architecture, but it cannot guarantee the robustness to different architectures. In terms of theoretical insights, the authors said this work is not certified defense. Even empirical methods could provide some theoretical insights to reveal why the proposed method works.
5. The authors admitted the incomplete reported results.

I would like to thank all professional comments and rebuttals from reviewers and authors, and now we have a clear understanding of the strengths and limitations of this work. Since most above important concerns have not been well addressed, indicating the intrinsic drawbacks of the proposed method, my recommendation is reject.

**Additional Comments On Reviewer Discussion:**

The rebuttal and discussions, as well as their influences in the decision, have been summarized in the above metareview.

---

### Decision · Program_Chairs · 2025-01-22

Reject